# Biomechanical analysis of hip, knee, and ankle joint contact forces during squats in elite powerlifters

Alexander Pürzel[1,2,3]*, Paul Kaufmann[1,3,4], Willi Koller[1,3], Elias Kaj Wallnöfer[1,2,3], Arnold Baca[1], Hans Kainz[1,3]

1 Neuromechanics Research Group, University of Vienna, Vienna, Austria, 2 Vienna Doctoral School of Pharmaceutical, Nutritional and Sport Sciences, University of Vienna, Vienna, Austria, 3 Centre for Sport Science and University Sports, University of Vienna, Vienna, Austria, 4 Université Côte d'Azur, LAMHESS, Nice, France

* alexander.puerzel@univie.ac.at

## Abstract

The squat is one of three lifts within the sport of powerlifting. This study examined how increasing intensity in the squat affects joint contact forces in elite powerlifters. Twenty-nine Austrian top-ranked powerlifters (16 male, 13 female) performed squats at 70% to 90% of their one-repetition maximum (1-RM). 3D motion capture and force plate data were used to estimate joint contact forces using musculoskeletal modelling. In contrast to the hip and ankle joints, which exhibited peak resultant joint contact forces in the deepest squat positions, the tibiofemoral and patellofemoral joints maintained consistently high loads over a broad portion of the squat cycle. During large parts of the concentric phase, the resultant joint contact forces did not significantly differ between intensity conditions, with the exception of the hip joint contact force. At 90% 1-RM, average peak joint contact forces reached $15.5 \pm 3.0$ times body weight (BW), $23.2 \pm 3.9$ BW, $26.7 \pm 4.3$ BW, and $11.5 \pm 2.2$ BW for the hip, tibiofemoral, patellofemoral, and ankle joint, respectively. The high and sustained joint contact forces observed in our study emphasise the need for load management strategies to optimise performance and reduce injury risk. These insights offer a valuable foundation for tailoring strength training programs and supporting long-term joint health in high-performance athletes.

## Introduction

In the sport of powerlifting, the squat represents the initial of three exercises, followed by the bench press and the deadlift. The objective is to achieve the highest possible weight in each of the lifts. Biomechanical and training science studies offer practical relevant insights in the squat for coaches and athletes alike. In addition to the advancement of knowledge concerning muscle forces [1–3], optimal technique [4–7] and

**Data availability statement:** All relevant data are within the manuscript and its Supporting Information files.

**Funding:** Open access funding is provided by the University of Vienna. The funders had no role in study design, data collection and analysis, decision to publish, or preparation of the manuscript.

**Competing interests:** The authors have declared that no competing interests exist.

training planning [8,9], it is imperative to recognise that squats can exert considerable forces on the involved joints – without [10–12] and with [13–15] additional external load.

In recent years, there has been a notable increase in the number of musculoskeletal modelling studies analysing joint contact forces during squats [16,17]. Simulations of bodyweight squats estimated tibiofemoral joint contact forces of $2.5 \pm 0.3$ body weight (BW) [17] and hip joint contact forces ranging from 2.5 to 3.0 BW [18]. In contrast, single-leg squats produced significantly higher hip joint contact forces, reaching $8.4 \pm 0.4$ BW [16]. Squats performed with an additional weight of $53 \pm 12$ kg resulted in patellofemoral joint contact forces between 6.0 BW and 7.1 BW [15].

Despite these recent insights, joint contact forces are often overlooked in the development and assessment of training programmes. However, if joint contact forces during exercises are beyond the physiological limits of joint structures, the training might exacerbate existing joint pathologies or even result in serious injuries. While the exact magnitude of force that is detrimental to joint structures, including ligaments, menisci and articular cartilage, remains uncertain [13,19], excessive stress on these structures can result in degenerative changes [20,21]. Conversely, appropriate compressive forces in the tibiofemoral joint may exert a hypertrophic effect on the cartilage [22]. The precise level of the joint contact forces and other load variables necessary for this effect remain to be determined through further research. Furthermore, compressive forces are of paramount importance for stabilising the knee joint, as they counteract anterior-posterior shear forces and, therefore, reduce the anterior translation of the tibia in relation to the femur while also contributing to frontal plane stability by limiting varus and valgus motion [23].

Worth noting, no study quantified joint contact forces during squats in powerlifting – particularly in a cohort of elite powerlifters with external barbell loads beyond 2 times BW. In order to enhance injury prevention, performance as well as rehabilitation, it is imperative for coaches and athletes to know how high joint contact forces are during squats. A deeper understanding of the magnitude and timing of joint contact forces during squats is essential for informed training planning, as these forces directly influence the mechanical loading of passive joint structures such as cartilage – thereby influencing their long-term adaptation, integrity, or degeneration.

The aim of this study was to examine the influence of an increasing intensity from 70% to 90% of the one-repetition maximum (1-RM) in the squat on joint contact forces of the hip, knee, and ankle joints of elite powerlifters. Two hypotheses were subsequently proposed: (a) the peak joint contact forces in each anatomical direction (anterior-posterior, medio-lateral, vertical, resultant) of each joint (hip, tibiofemoral, patellofemoral, ankle) exhibit a significant increase with increasing intensity; and (b) the joint contact forces exhibit a significant increase during the whole squat movement (expressed as % of the squat cycle) of the analysed joints with increasing intensity.

## Materials and methods

### Participants

The study comprised a cohort of 29 healthy elite powerlifters (13 female, 16 male) spanning various weight classes, as detailed in Table 1. The participants were either

**Table 1. Participant descriptive statistics.** Wilks score and IPF-GL points (International Powerlifting Federation Good Lift points), as indicated in the table, are both metrics that quantify the relationship between body weight and performance. A higher value indicates a higher relative performance.

| Descriptive | Mean | SD |
|---|---|---|
| Body mass (kg) | 83.1 | 19.4 |
| Height (cm) | 171.0 | 10.1 |
| Age (years) | 26.1 | 5.4 |
| Years of powerlifting training | 8.4 | 4.1 |
| Relative squat performance (xBM) | 2.4 | 0.4 |
| Wilks score | 418.7 | 40.0 |
| IPF-GL points | 86.3 | 8.5 |

IPF-GL points = International Powerlifting Federation Good Lift points. BM = body mass. SD = standard deviation.

current members of the Austrian national powerlifting team, competing internationally in Western European, European, and/or World Championships (n = 14), or had achieved a top three ranking at an Austrian powerlifting championship between 2019 and 2022 (n = 15). Due to the focus on elite powerlifters, who represent a narrowly defined and difficult to access population, the sample size of 29 participants is consistent with or even exceeds that of previous musculoskeletal modelling studies in strength-trained individuals [1,3]. All participants were required to maintain an active membership of the Austrian Powerlifting Federation, which mandates a medical examination and were recruited between 1 July and 30 September 2022. They had no current or previous musculoskeletal injuries that would hinder their ability to perform the squat. The study was approved by the Research Ethics Committee of the authors' institution (Ethics Committee of the University of Vienna, Ref. No.: 00771). Prior to their participation in the research study, the participants were provided with comprehensive information regarding the procedures, potential benefits, and any associated risks. Subsequent to an explanation of the research details, each participant provided written informed consent, indicating a clear understanding of the information presented. Furthermore, they were instructed to refrain from engaging in strenuous lower-body exercises for a minimum of three days prior to the data collection.

### Three-dimensional motion capturing

A three-dimensional motion capture system comprising 12 cameras (Vicon Motion System, Oxford, UK) was used to capture the trajectories of 73 (static position) and 53 (squat movement) markers on the lower limbs, upper body, and barbell with a recording frequency of 200 Hz (Fig 1). The captured marker data was labelled and processed with a Butterworth fourth-order low-pass filter with a cutoff frequency of 6 Hz using Nexus 2.14.1 software (Vicon Motion System, Oxford, UK). The ground reaction forces were recorded simultaneously at a frequency of 1000 Hz via two separate force plates (Kistler Instrumente, Winterthur, Switzerland). During the squat, participants were permitted to utilise their own equipment, including footwear (with or without elevated heels), knee sleeves, powerlifting belts, and wrist wraps, provided that all equipment complied with the standards of the International Powerlifting Federation (IPF) [24]. The warm-up protocol, nutritional intake prior to and during the testing, and rest intervals between attempts were not subject to any restrictions, thereby enabling the participants to adequately prepare and perform in accordance with their accustomed routines. Each participant performed a single squat at 70%, 75%, 80%, 85%, and 90% of their 1-RM, in accordance with the technique standards of the IPF [25]. It was imperative that the hip joint descended below the knee joint at the lowest point of the squat. To guarantee the safety of the participants, three spotters were present for each squat, and verbal encouragement was provided throughout each attempt.

The 1-RM values were approximated by analysing the participants' recent training and competition results, incorporating feedback from both the participants and their coaches. This approach was adopted in response to concerns raised by

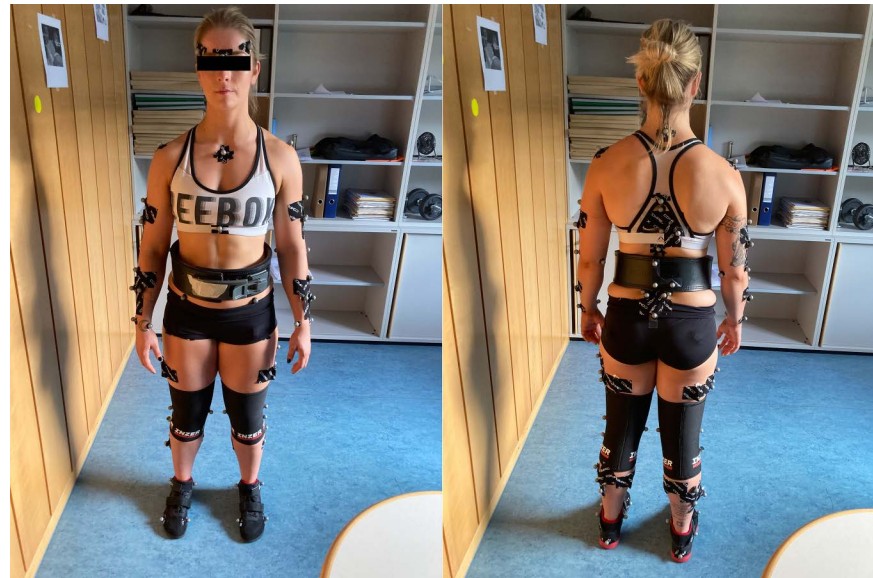

**Fig 1. Marker placement on the athletes.** temporal right/left (ri/le), glabella, acromion ri/le, manubrium, proc. xiphoideus, epicondylus lat. ri/le, epicondylus med. ri/le, ulna distalis ri/le, radius distalis ri/le, upper arm cluster ri/le, lower arm cluster ri/le, spina iliaca anterior superior (SIAS) ri/le, spina iliaca posterior superior (SIPS) ri/le, condylus med. ri/le, condylus lat. ri/le, caput fib. ri/le, malleolus lat. ri/le, malleolus med. ri/le, calcaneus ri/le, metatarsus I ri/le, metatarsus V ri/le, thigh cluster ri/le, shank cluster ri/le, cervical vertebra (C)7, thoracic vertebra (T)6, T12, upper back cluster, sacrum cluster, belt cluster, bar ri/le. 20 markers were removed after the static calibration trial. These markers include the epicondylus lat. ri/le, epicondylus med. ri/le, radius distalis ri/le, SIAS ri/le, SIPS ri/le, condylus med. ri/le, condylus lat. ri/le, caput fib. ri/le, malleolus lat. ri/le, and malleolus med. ri/le markers.

some participants and coaches that performing 1-RM tests might have a detrimental impact on their training and upcoming competition performance.

## Magnetic resonance imaging

In six of the 29 participants, magnetic resonance imaging (MRI) was conducted on the lower limbs using a 3 Tesla MRI scanner (MAGNETOM Vida, Siemens, Berlin/Munich, Germany) with a T1 vibe sequence, providing a voxel resolution of 0.98x0.98x1.0 mm, MRI images were used for assessing muscle volume. The MRI-based muscle volume data from these six participants was used to calculate a mean scaling factor for maximum isometric muscle force, which was then applied to all participants in our study (for details, see section "Calculation of joint contact forces"). Muscle segmentation was performed using the open-source software 3D Slicer (version 5.2.2) [26]. Manual annotations were created with at least ten slices per muscle marked in each anatomical plane to provide a robust basis for interpolation. The remaining layers were interpolated, generating continuous 3D segmentations of the target muscles (mm. vasti, m. rectus femoris, m. semimembranosus, m. gluteus maximus). If interpolation errors were detected, additional slices were manually segmented until an anatomically accurate muscle representation was achieved and final muscle volumes were calculated. For a better illustration of the process, please refer to S5 Fig in the Supporting Information.

## Calculation of joint contact forces

Musculoskeletal modelling in OpenSim 4.4 [27] was used to estimate joint contact forces. All calculations were based on the "Catelli"-model [28], a validated model for high hip and knee flexion. The initial stage of the process involved scaling the base model to the participants' anthropometry based on the location of the surface markers [29]. In order to ensure

that the model is capable of producing necessary internal joint moments to resist the external moments, the maximum isometric muscle forces ($F$) for all models were adjusted based on their body mass ($M$) using Equation 1 [30,31].

$$F_{iso,scaled} = F_{iso,generic} * \left( \frac{M_{scaled}}{M_{generic}} \right)^{\frac{2}{3}}$$

(1)

Despite this muscle force scaling process, the simulations resulted in reserve actuator forces that were beyond 10% of the net joint moments, thereby producing simulations that were not realistic [32]. In other words, the models were too weak to perform the high load exercises of the elite-level athletes. In order to obtain a more precise estimation of the isometric maximum force of the participants and to make the necessary adjustments to the models, MRI scans of the lower limbs were analysed from six participants. The mm. vasti, m. rectus femoris, m. semimembranosus, and m. gluteus maximus muscles were segmented and their volumes were calculated using the 3D Slicer 5.2.2 [26]. The maximum isometric muscle force was subsequently calculated using Equation 2 [33,34], where $F_{iso}$ represents the muscle's isometric maximum force, $ss$ means subject specific, $V$ is the muscle volume determined by the MRI scans and $l_o^m$ is the optimal fibre length (OpenSim 4.4 default [27]). The maximum fibre tension ($\sigma$) was set to 105 N cm$^{-2}$, a value which falls within the range previously reported in the literature [35]. This choice was based on anatomical cross-sectional areas measured in body-builders [36] and was adjusted for muscle pennation angle [36–38] to better reflect the physiological cross-sectional area.

$$F_{iso,ss} = \frac{V_{ss}}{l_{o,ss}^m} * \sigma$$

(2)

The maximum isometric force calculated using Equation 1 was found to be three times lower ($S$ obtained with Equation 3) than the corresponding estimate using Equation 2, which was based on MRI-derived muscle volume data.

$$S \approx mean \frac{F_{iso,ss}}{F_{iso,scaled}}$$

(3)

Consequently, the final equation (Equation 4) employed for the scaling of isometric forces across all 29 models involved multiplying the initial equation (Equation 1) by factor $S$ (Equation 3), to ensure that the models are capable of withstanding the external loads.

$$F_{iso,scaled\_final} = F_{iso,generic} * \left( \frac{M_{scaled}}{M_{generic}} \right)^{\frac{2}{3}} * S$$

(4)

Although Equation 2 was applied only to the muscle volume values of the MRI subgroup (6 athletes), its output was used to define the scaling factor in Equation 4. Given the homogeneity of the participants with respect to their performance level, Equation 4 with $S = 3$ was employed to scale the isometric maximum force for the models of all 29 participants.

After scaling the models to the anthropometry of each participant [29], inverse kinematics and inverse dynamics was employed to calculate joint angles and joint moments, respectively. Subsequently, Static Optimization was employed to estimate muscle forces and activations by minimising the sum of squared muscle activations [27]. Ultimately, Joint Reaction Analysis was conducted to determine the hip, tibiofemoral, patellofemoral, and ankle joint contact forces, considering contributions from muscle forces [39,40], as well as the external loads. Hip joint contact forces and orientations are reported in relation to the local coordinate system of the femur. Tibiofemoral contact forces and orientations are reported in relation to the local coordinate system of the tibia. Patellofemoral contact forces and orientations are expressed in relation to the local coordinate system of the patella. Ankle joint contact forces and orientations are reported in relation to the local coordinate system of the talus.

## Model validation

To validate our simulation results, a qualitative visual comparison of the hip and knee joint contact forces calculated during unloaded squats of one athlete was made with those found on OrthoLoad [41], a public database of hip and knee joint contact forces measured in vivo with instrumented hip and knee implants. Furthermore, for the same participant, electromyography (EMG) data were available and used as an additional validation step for evaluating the plausibility of the simulated neuromuscular control patterns during squatting. For the 70% 1-RM condition, we qualitatively compared EMG signals of m. biceps femoris, m. gluteus maximus, m. rectus femoris, and m. vastus lateralis with muscle activation output from our musculoskeletal simulations (i.e., obtained from Static Optimization).

## Data analyses

The squat cycle in this study was defined as follows: The cycle began when the C7 marker (placed on the seventh cervical vertebra) descended by 2% of its initial vertical distance to the floor, marking 0% of the squat cycle. The lowest point of the squat (50% of the cycle) corresponded to the minimum vertical position of the sacrum marker. The cycle was considered complete (100%) when the C7 marker returned to 98% of its initial vertical height. Joint contact force waveforms were normalized over the squat cycle to 101 data points. The eccentric (0% – 50%) and concentric phases (51% to 100%) were analysed separately and averaged across both legs. Peak joint contact forces were identified for each phase individually.

To detect changes in joint contact force waveforms throughout the squat cycle across different intensity conditions (70%, 75%, 80%, 85%, 90% of 1-RM), Statistical Parametric Mapping (SPM) [42] was used, utilising the SPM1D package [43] in Matlab R2022a (Mathworks Inc., Natick, MA, USA). In instances where a statistically significant result was obtained by the repeated measures ANOVA ($p \leq 0.05$), a Bonferroni correction was applied for post-hoc pairwise comparisons.

A two-way repeated measures ANOVA was conducted to evaluate the effects of phases (eccentric and concentric) and intensities (70% to 90% of 1-RM) on peak joint contact forces for each joint and anatomical direction. Subsequently, a Bonferroni correction was applied for the purposes of conducting post-hoc pairwise comparisons. This statistical analysis was conducted using JASP (version 0.18.1.0), with a significance level of $p \leq 0.05$. A post-hoc power analysis (ANOVA: repeated measures, within factors) was conducted using G*Power (version 3.1.9.7) [44], yielding a result of 93.5%.

## Results

### Validation of simulation results

The shape of the hip and knee joint contact force waveforms from our simulations showed a reasonable agreement with the values from instrumented implants (S1 Fig in the Supporting Information) but with higher force magnitudes. Moreover, hip joint contact forces showed similar shape compared to previous reported values from a musculoskeletal modelling study [16] (S2 Fig in the Supporting Information). Estimated muscle activations from Static Optimization showed reasonable agreement with experimental EMG signals for all analysed muscles, except the m. rectus femoris (S3 Fig in the Supporting Information).

### Peak joint contact forces

With the exception of the medio-lateral force component, the peak joint contact forces increased with increasing intensities in all joints and in all anatomical direction analysed (Fig 2). With regard to the phase of the squat cycle (eccentric, concentric), peak hip, tibiofemoral, and ankle joint contact forces were significantly higher in the concentric phase than in the eccentric phase. Patellofemoral joint contact forces did not show any significant difference related to the phase of the squat cycle.

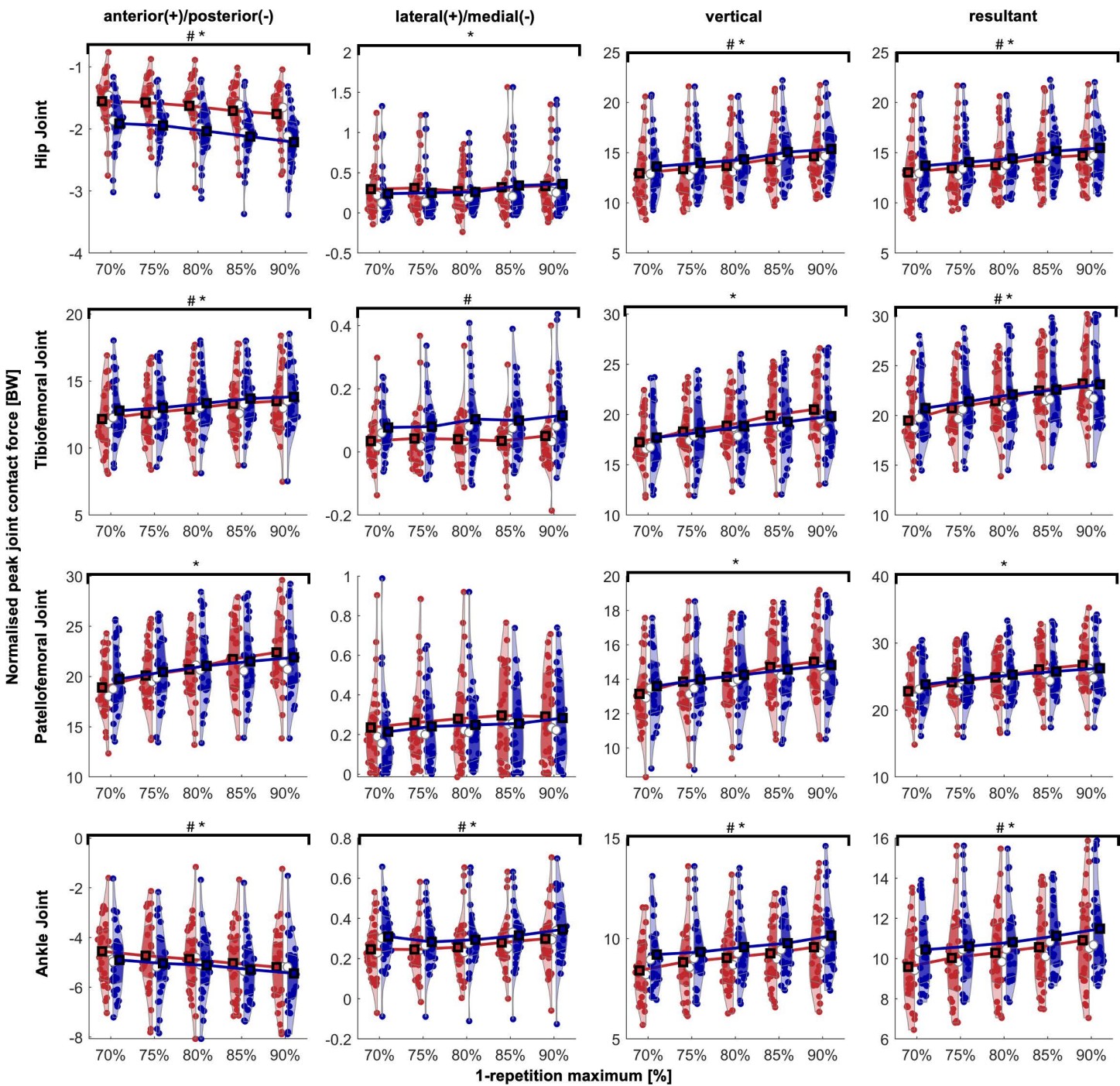

**Fig 2. Peak joint contact forces during squats with different intensities.** Each coloured circle represents one participant; black squares represent mean values; white circles indicate median values. Shaded areas represent violin plots, illustrating the distribution of data; darker areas indicate quartiles, while lighter areas show the remainder of the distribution; red is eccentric, blue is concentric; # indicates significant difference for phase (eccentric, concentric); * indicates significant difference for intensity.

The highest joint contact forces occurred at 90% 1-RM for each joint. Based on the mean values at 90% 1-RM across all participants, the peak resultant joint contact forces were 15.5±3.0 BW, 23.2±3.9 BW, 26.7±4.3 BW, and 11.5±2.2 BW for the hip, tibiofemoral, patellofemoral and ankle joint, respectively. Detailed values for joint contact forces in all anatomical directions, along with the exact statistical significance level ($p$), including the effect size ($\eta^2_p$) and the percentage change relative to 70% 1-RM of the corresponding phase, can be found in Table 2.

## Joint contact force waveforms

Across a broad spectrum of the squat cycle, joint contact forces significantly changed with an increasing intensity in both the eccentric and concentric phases (Fig 3). However, there were also specific sections of the squat cycle where, despite increasing intensity, there was no significant change.

At the hip joint, the anterior-posterior force component was oriented in a posterior direction (posteriorly oriented force on femur in relation to pelvis) and demonstrated a rise in magnitude with increasing intensity. The medio-lateral forces, which showed only minimal changes with increasing intensities, were small compared to the other force components and had a medially oriented vector. A significant increase in vertical hip joint contact force was observed over almost the entire squat cycle with increasing intensity conditions. The resulting hip joint contact force exhibited a pattern comparable to that observed for the vertical forces.

In the tibiofemoral joint, the anterior-posterior, the vertical and the resultant force significantly increased with increasing intensities for most parts of the squat cycle. Similar to the hip joint, the media-lateral forces were relatively small compared to the other force components. In contrast to the anterior-posterior forces, which were highest around 50% of the squat cycle, the vertical and resultant forces showed two peaks – one in the middle of the eccentric phase and one in the middle of the concentric phase.

At the patellofemoral joint, the medio-lateral forces did not change across the intensity levels and were low compared to the other anatomical directions. In the remaining directions, significant differences between the intensity conditions were observed over parts of the squat cycle. However, the early eccentric phase and large parts of the concentric phase did not show any significant differences with increasing intensities.

The ankle joint exhibited a maximum contact force around 50% of the squat cycle. Significant differences with increasing intensity in all anatomical force directions, particularly in the eccentric phase of the squat, were observed. The medio-lateral, vertical and resultant ankle joint contact forces barely changed during the concentric phase with increasing intensities.

## Discussion

We showed that powerlifters are subjected to considerable joint contact forces during the squat, with some participants in our study experiencing joint contact forces beyond 20 times their BW. Our first hypothesis – peak joint contact forces increase with increasing intensities – was largely confirmed by our results. Over long periods of the squat cycle, joint contact forces increased in all joints and most anatomical force directions. Our second hypothesis – joint contact forces exhibit an increase throughout the squat cycle with increasing intensities – could only be partly confirmed. Interestingly, during substantial parts of the concentric phase, the tibiofemoral, patellofemoral and ankle joint contact forces showed no significant difference between intensity conditions. One possible explanation for this could be that muscle forces in this region did not significantly increase as intensity increased, as shown in S4 Fig in the Supporting Information. Since muscle forces play a crucial role in generating joint contact forces [39,40], constant muscle forces – as seen in some but not all muscles [2,45] – might explain the relatively constant joint contact forces during the concentric phase.

Peak hip and ankle joint contact forces were higher in the concentric phase than in the eccentric phase. This might be partly due to the muscle's force-velocity properties. Muscle fibers can produce greater active tension at slower shortening velocities and at optimum lengths [46]. During the upward phase of the squat, which is usually performed with a very slow

Table 2. Peak joint forces [BW in mean±SD] and percentage change relative to 70% 1-RM of the corresponding phase in the hip, knee, and ankle joint with increasing intensity (70%, 75%, 80%, 85%, 90% of 1-RM).

| Joint | Direction of force | 70% 1-RM Ecc | 70% 1-RM Con | 75% 1-RM Ecc | 75% 1-RM Con | 80% 1-RM Ecc | 80% 1-RM Con | 85% 1-RM Ecc | 85% 1-RM Con | 90% 1-RM Ecc | 90% 1-RM Con | Result for phase | Result for intensity |
|---|---|---|---|---|---|---|---|---|---|---|---|---|---|
| Hip | ap | −1.55±0.34 | −1.91±0.42 | −1.57±0.38 (1.3±2.5%) | −1.95±0.40 (2.1±2.9%) | −1.63±0.44 (5.2±3.4%) | −2.04±0.46 (6.8±3.0%) | −1.71±0.40 (10.3±3.1%) | −2.13±0.45 (11.5±3.0%) | −1.76±0.42 (13.5±3.3%) | −2.21±0.48 (15.7±3.2%) | Con>Ecc (absolute value) ($p<0.001$*, $\eta^2 p=0.513$) | Increases in negative direction with intensity ($p<0.001$*, $\eta_p^2=0.113$) |
| | ml | 0.30±0.33 | 0.24±0.34 | 0.31±0.34 (3.3±15.8%) | 0.25±0.34 (4.2±19.9%) | 0.27±0.28 (−10.0±14.5%) | 0.26±0.26 (8.3±17.0%) | 0.32±0.39 (6.7±17.5%) | 0.34±0.39 (41.7±22.2%) | 0.34±0.37 (13.3±17.3%) | 0.36±0.41 (50.0±22.6%) | Con=Ecc ($p=0.632$; $\eta_p^2=0.003$) | Increases with intensity ($p=0.004$*, $\eta^2=0.059$) |
| | v | 12.95±3.20 | 13.63±3.07 | 13.37±3.28 (3.2±4.2%) | 13.98±3.06 (2.6±3.4%) | 13.63±3.03 (5.3±4.0%) | 14.33±2.98 (5.1±3.3%) | 14.37±3.12 (11.0±4.0%) | 15.05±3.05 (10.4±3.3%) | 14.65±3.05 (13.1±4.1%) | 15.36±2.97 (12.7±3.3%) | Con>Ecc ($p<0.001$*, $\eta_p^2=0.122$) | Increases with intensity ($p<0.001$*, $\eta^2=0.450$) |
| | r | 13.03±3.21 | 13.70±3.08 | 13.45±3.29 (3.2±4.3%) | 14.06±3.07 (2.6±3.3%) | 13.77±3.03 (5.7±4.0%) | 14.42±2.98 (5.3±3.2%) | 14.46±3.12 (11.0±4.0%) | 15.16±3.05 (10.7±3.2%) | 14.73±3.05 (13.1±4.1%) | 15.46±2.97 (12.8±3.3%) | Con>Ecc ($p<0.001$*, $\eta_p^2=0.123$) | Increases with intensity ($p<0.001$*, $\eta^2=0.449$) |
| Tibio-femoral | ap | 12.17±2.38 | 12.79±2.35 | 12.59±2.35 (3.5±3.2%) | 13.02±2.27 (1.8±2.7%) | 12.91±2.46 (6.1±3.3%) | 13.35±2.41 (4.4±2.8%) | 13.31±2.42 (9.4±3.3%) | 13.69±2.32 (7.0±2.8%) | 13.50±2.50 (10.9±3.4%) | 13.82±2.44 (8.0±2.9%) | Con>Ecc ($p<0.001$*, $\eta_p^2=0.129$) | Increases with intensity ($p<0.001$*, $\eta^2=0.512$) |
| | ml | 0.03±0.09 | 0.08±0.08 | 0.04±0.10 (33.3±353%) | 0.08±0.10 (0.0±176%) | 0.04±0.09 (33.3±300%) | 0.10±0.13 (25.0±229%) | 0.04±0.08 (33.3±267%) | 0.10±0.11 (25.0±190%) | 0.05±0.11 (66.7±367%) | 0.12±0.13 (50.0±229%) | Con>Ecc ($p=0.005$*, $\eta_p^2=0.148$) | Stays constant with intensity ($p=0.079$; $\eta_p^2=0.017$) |
| | v | 17.27±2.77 | 17.72±3.36 | 18.31±2.97 (6.0±3.0%) | 18.23±3.33 (2.9±3.4%) | 18.89±2.99 (9.4±3.0%) | 18.84±3.84 (6.3±5.5%) | 19.90±3.40 (15.2±3.6%) | 19.28±3.94 (8.8±6.2%) | 20.51±3.49 (18.8±3.8%) | 19.85±3.94 (12.0±6.4%) | Con=Ecc ($p=0.396$; $\eta_p^2=0.005$) | Increases with intensity ($p<0.001$*, $\eta^2=0.480$) |
| | r | 19.50±3.10 | 20.74±3.75 | 20.75±3.34 (6.4±4.6%) | 21.41±3.69 (3.2±4.3%) | 21.46±3.33 (9.5±4.6%) | 22.13±4.14 (6.7±6.1%) | 22.49±3.68 (15.3±5.7%) | 22.59±4.13 (9.0±6.5%) | 23.22±3.86 (19.3±6.2%) | 23.13±4.35 (11.6±6.8%) | Con>Ecc ($p=0.021$*, $\eta_p^2=0.031$) | Increases with intensity ($p<0.001$*, $\eta^2=0.531$) |

*(Continued)*

**Table 2.** (Continued)

| Joint | Direction of force | 70% 1-RM Ecc | 70% 1-RM Con | 75% 1-RM Ecc | 75% 1-RM Con | 80% 1-RM Ecc | 80% 1-RM Con | 85% 1-RM Ecc | 85% 1-RM Con | 90% 1-RM Ecc | 90% 1-RM Con | Result for phase | Result for intensity |
|---|---|---|---|---|---|---|---|---|---|---|---|---|---|
| Patello-femoral | ap | 18.92±3.03 | 19.77±3.51 | 20.11±3.24 (6.3±3.5%) | 20.45±3.51 (3.4±5.0%) | 20.72±3.26 (9.5±3.6%) | 21.07±3.90 (6.6±5.5%) | 21.74±3.65 (14.9±4.3%) | 21.51±3.90 (8.8±5.5%) | 22.40±3.80 (18.4±4.6%) | 21.90±4.15 (10.8±5.7%) | Con=Ecc ($p=0.396$; $\eta^2_p=0.004$) | Increases with intensity ($p<0.001$*; $\eta^2_p=0.533$) |
| | ml | 0.24±0.21 | 0.21±0.22 | 0.26±0.22 (8.3±12.7%) | 0.24±0.19 (14.3±14.3%) | 0.28±0.24 (16.7±14.2%) | 0.25±0.22 (19.0±10.7%) | 0.30±0.24 (25±14.2%) | 0.26±0.21 (23.8±13.8%) | 0.29±0.23 (20.8±13.5%) | 0.28±0.23 (33.3±14.6%) | Con=Ecc ($p=0.092$; $\eta^2_p=0.016$) | Stays constant with intensity ($p=0.07$; $\eta^2_p=0.051$) |
| | v | 13.15±2.16 | 13.60±2.14 | 13.86±2.24 (5.4±3.0%) | 14.00±2.29 (2.9±3.0%) | 14.14±2.11 (7.5±2.8%) | 14.26±2.28 (4.9±2.9%) | 14.71±2.16 (11.9±2.7%) | 14.57±2.22 (7.1±2.8%) | 15.02±2.32 (14.2±2.9%) | 14.82±2.31 (8.9±2.9%) | Con=Ecc ($p=0.342$; $\eta^2_p=0.003$) | Increases with intensity ($p<0.001$*; $\eta^2_p=0.591$) |
| | r | 22.80±3.57 | 23.83±3.95 | 24.17±3.82 (4.8±4.4%) | 24.61±4.00 (3.3±2.4%) | 24.88±3.75 (5.3±3.8%) | 25.27±4.35 (6.0±2.8%) | 26.07±4.11 (10.9±4.4%) | 25.77±4.31 (8.1±2.8%) | 26.73±4.30 (12.9±4.0%) | 26.25±4.64 (10.2±3.2%) | Con=Ecc ($p=0.279$; $\eta^2_p=0.005$) | Increases with intensity ($p<0.001$*; $\eta^2_p=0.559$) |
| Ankle | ap | -4.55±1.31 | -4.90±1.32 | -4.73±1.47 (-4.0±5.0%) | -5.04±1.43 (-2.9±4.5%) | -4.87±1.42 (-7.0±4.8%) | -5.12±1.36 (-4.5±4.2%) | -5.03±1.37 (-10.5±4.7%) | -5.29±1.33 (-8.0±4.1%) | -5.19±1.55 (-14.1±5.3%) | -5.44±1.48 (-11.0±4.4%) | Con>Ecc (absolute value) ($p<0.001$*; $\eta^2_p=0.128$) | Increases in negative direction with intensity ($p=0.001$; $\eta^2_p=0.270$) |
| | ml | 0.25±0.14 | 0.31±0.15 | 0.25±0.14 (0.0±7.8%) | 0.28±0.14 (-9.7±6.8%) | 0.26±0.15 (4.0±8.0%) | 0.29±0.16 (-6.5±7.4%) | 0.28±0.14 (12.0±7.8%) | 0.32±0.15 (3.2±6.8%) | 0.30±0.16 (20.0±8.9%) | 0.35±0.17 (12.9±7.2%) | Con>Ecc ($p=0.002$*; $\eta^2_p<0.104$) | Increases with intensity ($p<0.001$*; $\eta^2_p=0.089$) |
| | v | 8.41±1.56 | 9.20±1.71 | 8.82±1.92 (4.9±6.0%) | 9.33±1.78 (1.4±5.5%) | 9.03±1.78 (7.4±5.5%) | 9.58±1.72 (4.1±5.2%) | 9.26±1.67 (10.1±5.2%) | 9.77±1.48 (6.2±4.2%) | 9.57±2.01 (13.8±6.0%) | 10.14±1.92 (10.2±5.2%) | Con>Ecc ($p<0.001$*; $\eta^2_p=0.135$) | Increases with intensity ($p<0.001$*; $\eta^2_p=0.204$) |
| | r | 9.59±1.93 | 10.44±1.92 | 10.03±2.33 (4.6±6.3%) | 10.61±2.13 (1.6±5.4%) | 10.29±2.16 (7.3±5.7%) | 10.82±2.00 (3.6±5.1%) | 10.57±2.03 (10.2±5.4%) | 11.13±1.76 (6.6±4.3%) | 10.91±2.43 (13.7±6.3%) | 11.48±2.21 (10.0±5.3%) | Con>Ecc ($p<0.001$*; $\eta^2_p=0.137$) | Increases with intensity ($p<0.001$*; $\eta^2_p=0.239$) |

1-RM = 1-repetition maximum; ap = anterior-posterior; v = vertical; r = resultant; ecc = eccentric phase; con = concentric phase; * indicates significance.

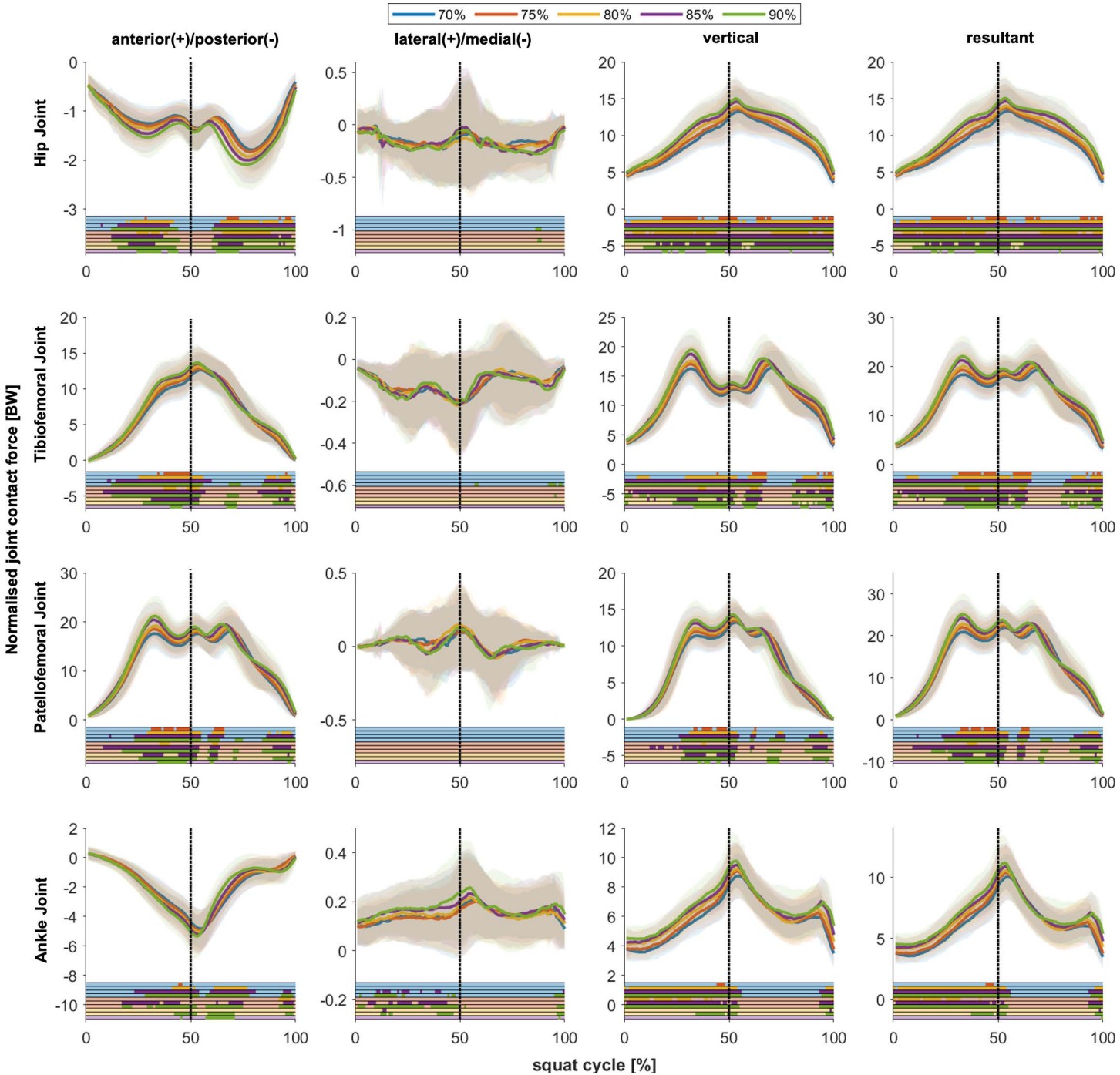

**Fig 3. Joint contact forces throughout the squat cycle with different intensities.** 0% and 100% of the squat cycle represent the respective upright position, and 50% denotes the lowest point of the sacrum, thus representing the deepest squat position. Solid lines in the figures represent the mean joint contact force across participants. Shaded areas represent ± one standard deviation. The lines below the figures indicate significant differences (SPM) during the squat cycle. The first four lines (light blue) represent significant differences between 70% 1-RM and 75% 1-RM, 70% 1-RM and 80% 1-RM, 70% 1-RM and 85% 1-RM, and 70% 1-RM and 90% 1-RM. The next three lines (orange) represent significant differences between 75% 1-RM and 80% 1-RM, 75% 1-RM and 85% 1-RM, and 75% 1-RM and 90% 1-RM. The following two lines (yellow) show significant difference between 80% 1-RM and 85% 1-RM, and 80% 1-RM and 90% 1-RM The last line (purple) shows significant differences between 85% 1-RM and 90% 1-RM.

velocity, the muscles shorten at relatively low velocities – approaching near-isometric conditions – which enables the generation of very high muscle forces. These forces are necessary to overcome the combined effects of barbell weight, inertia, and gravity. In fact, it is precisely the necessity to actively counteract gravity during the upward phase – rather than "following it", as in the downward phase – that increases the muscular demand, which in turn leads to slower contraction velocities, higher muscle forces and, consequently, elevated joint contact forces [39,40].

Both eccentric and concentric training improved function and reduced pain in patients with symptomatic knee osteoarthritis (OA) [47]. A systematic review showed that eccentric training may offer potential benefits in musculoskeletal rehabilitation [48]. We found lower hip and ankle joint contact forces in the eccentric compared to the concentric phase. However, it is important to note that the differences of the resultant joint contact forces between the eccentric and concentric phase were consistently below 10% BW (S1 Table). While these differences are small, they could still be of potential relevance in specific contexts, such as injury prevention and rehabilitation.

Comparing our findings with previous research is difficult due to the absence of studies with elite powerlifters squatting with high barbell loads. In the field of performance-oriented squats, Salem et al. examined the relationship between joint contact forces and squat depth, which showed that patellofemoral joint contact forces only increased slightly from 6.0 BW to 7.1 BW with higher knee flexion [15]. It is important to note that although the participants in the study conducted by Salem et al. performed squats with 85% 1-RM, the additional load utilized was on average only $53 \pm 12$ kg [15], which is a lot lower compared to our participants with an average barbell load of $180 \pm 61$ kg at the 90% 1-RM condition and likely explains the higher forces in our study (detailed information on the 1-RM values of our participants is available in S2 Table of the Supporting Information).

In vivo measured joint contact forces using instrumented endoprostheses ranged from 2.1 to 6.5 BW in the patellofemoral joint [49] and from 1.6 to 5.4 BW in the tibiofemoral joint [49]. In the hip joint, forces of approximately 2 BW were observed during squats without additional loads [50]. Musculoskeletal simulations of body weight squat estimated $2.5 \pm 0.3$ BW for the tibiofemoral joint [17] and 2.5 to 3.0 BW for the hip joint [18]. When considering that all of the aforementioned values represent squats without additional loads, that our athletes lifted far more than twice their BW as additional load on the barbell in a deep squat movement, and that our estimated joint contact force waveforms exhibit a similar shape to those measured by instrumented endoprostheses, we are confident that our values are reasonable.

A factor not considered in this study, but which could have an influence on joint contact forces, is the contact between the calf and the thigh. This contact can result in substantial force transfer during squats [51]. Both compression and shear forces in the knee joint are reduced by the thigh-calf contact during high knee flexion [52,53]. This effect is more pronounced the greater the sum of the thigh and calf circumferences [53]. As our participants were elite powerlifters, this effect is likely to be important due to their large muscle mass and should be taken into account in future studies as the influence on muscle and joint contact forces could be substantial. The use of knee sleeves worn by all participants could have further enhanced this effect [54]. Similarly, contact between the trunk and thighs and the wearing of a belt can also lead to changes in joint contact forces. Unfortunately, there are no studies on this aspect yet.

The individual morphological characteristics may also influence joint contact forces and the resulting stress on the joint structures. It is important to note that the joint contact forces represent only one factor among many that contribute to the overall stress experienced by joint structures. The dimensions and mechanical characteristics of the joint surfaces and surrounding tissue influence the stress on the anatomical structures such as cartilage and ligaments. To illustrate, identical joint contact forces but different femoral head sizes can result in different hip cartilage stress. Hence, the individual-specific bony geometry [55,56], additionally to the movement execution of the squat [49,57], determine the stress placed on the involved anatomical structures.

Heavy resistance training can elicit both adaptive and maladaptive responses in articular cartilage. On the one hand, high mechanical loading may promote functional cartilage adaptation, as elite weightlifters show greater knee cartilage thickness than non-athletes, suggesting post-maturation hypertrophy [58,59]. Moderate exercise can also enhance

cartilage composition [60]. Consequently, long-term supervised strength training appears to be safe for cartilage and may be protective against OA [61]. Animal studies further indicate that specific high-impact stimuli, like plyometric training, can increase cartilage thickness [62].

Conversely, excessive or prolonged joint loading may contribute to cartilage degeneration. Epidemiological data show elevated rates of knee OA, particularly in the patellofemoral joint, among former strength athletes [63]. A recent systematic review further links weightlifting and other high-impact sports to molecular and structural cartilage changes conducive to OA [64]. Elevated body mass may further exacerbate joint stress, partly explaining the increased OA risk in weightlifters [63]. These findings support the concept of a loading threshold for cartilage homeostasis, beyond which catabolic processes may be triggered [64]. The joint contact force data presented in this study provide context for these findings, although the determination of individualised load thresholds remains a topic for future investigation.

In light of the capacity for both adaptive and maladaptive cartilage responses to heavy resistance training, appropriate load management is essential to maximise benefits and minimise risks in strength sports. While mechanical loading can enhance cartilage properties, avoiding chronic overload is essential. Principles such as gradual progression, adequate recovery, and targeted injury prevention are crucial to support joint health. Hence, with well-designed training protocols, cartilage adaptation may be facilitated, while unrelenting stress may lead to degeneration. Although deep squats generate high joint contact forces, they are not inherently detrimental when executed with proper technique and progressive loading [65]. A nuanced understanding of joint mechanics and periodisation strategies – including strategic deload phases [66,67] – is critical to balancing performance goals with joint preservation.

While musculoskeletal simulations using Static Optimization offer computational efficiency and simplicity, it has inherent limitations that can affect the accuracy of joint contact force predictions. Static Optimization operates under the assumption that muscle forces can be determined independently at each time point, disregarding the temporal dynamics of muscle activation and contraction [68]. Further, it typically aims to minimise a cost function, such as the sum of squared muscle activations, which may lead to an underestimation of muscle co-contraction. This can result in inaccuracies in joint contact force estimations, particularly in movements where co-contraction plays a role [69]. As illustrated in the validation plot (S3 Fig in the Supporting Information), the activation waveforms estimated via Static Optimization showed a good agreement with surface EMG signals across most muscles. However, activation of the rectus femoris was virtually absent in the simulation, despite clear EMG activity in this muscle. This discrepancy is a known and well-documented limitation of Static Optimization approaches, particularly for biarticular muscles such as the rectus femoris [70–72]. The minimisation-based cost function used in Static Optimization tends to reduce activations that generate opposing joint torques, which can result in physiologically unrealistic suppression of such muscles during tasks involving simultaneous hip and knee extension. Incorporating participant-specific electromyography data in the simulations might enhance the accuracy of muscle force estimations.

This study had some limitations. First, participants performed only one squat per intensity level. Given their extensive training and the large sample size, we consider this sufficient to test our hypotheses. Moreover, the inclusion of additional repetitions at the high intensity levels would have introduced a bias due to fatigue. Secondly, the influence of individual differences in bone structure, muscle attachment points and, consequently, moment arms [56] and individual muscle coordination strategies (i.e., EMG-informed neuromusculoskeletal modelling) [73–75] was not considered. However, previous studies have shown that muscle forces – and consequently, joint contact forces – are more significantly affected by movement patterns than by bone morphology [76], while bone morphology itself has a greater effect than muscle coordination [77]. Therefore, the most critical factors were considered in our study. Likewise, applying finite element (FE) modelling would enable the evaluation of tissue-level loads (cartilage stress, tendon strain, subchondral bone pressure). For example, FE analyses of deep squatting reveal markedly elevated articular cartilage stresses relative to neutral standing [78], a finding that underscores the need to assess such high-load scenarios to understand potential tissue damage and long-term joint adaptation. Future research should include advanced EMG- and medical imaging-informed musculoskeletal

simulations to enhance participant-specific estimates of joint contact forces as well as FE simulations to quantify tissue-level loads in elite powerlifters. Thirdly, the validation of the simulation was based on in vivo data from individuals with instrumented joint replacements [41]. These datasets originate from an older and less athletic population with substantially lower external loads than those observed in our elite powerlifters. Moreover, Schellenberg et al. [79] showed that musculo-skeletal models can overestimate tibiofemoral joint contact forces by up to 60% at deep knee flexion angles (≥80°), which are common in powerlifting squats. However, their findings refer only to models that were available at the time of their study. In our work, we used a musculoskeletal model [28] that was published after Schellenberg et al. [79] and specifically designed to improve accuracy in deep flexion. Although direct in vivo validation in athletic populations remains challenging, this more recent model implementation may reduce the overestimation bias in high flexion angles. Nevertheless, the absolute joint contact force values – particularly in deep flexion – should still be interpreted with caution.

## Conclusion

The current study demonstrated that the joint contact forces of the hip, tibiofemoral, patellofemoral and ankle joint increased in squats with increasing intensities, yet not during all periods of the squat cycle. Joint contact forces in elite powerlifters reached more than 20 times their BW. These results have several practical implications. From a performance perspective, our study highlights the importance of load tolerance in the hip and knee joints. Given the high joint contact forces observed, coaches should consider structured periodisation and deloading strategies to balance adaptation with joint preservation. Variations in squat technique may also help modulate joint stress while maintaining stimulus. Furthermore, understanding joint contact force profiles can inform rehabilitation protocols to minimise joint loads. While the study focused on elite athletes, the insights may also apply to well-trained individuals or clinical populations seeking to build strength with preserving joint integrity over time. Knowledge of joint contact forces during squats can help to optimise load progression and movement selection, contributing to long-term joint health and injury prevention in both competitive and general strength training settings.

## Supporting information

**S1 Fig. Validation of simulations.** Hip and knee resultant joint contact force waveforms of squats (body weight only, 15 kg bar, 35 kg total) of one participant were comparable to those obtained from a participant with an instrumented hip implant and another participant with an instrumented knee implant from the Orthoload database. The shape of the Ortho-load waveforms were similar to the waveforms obtained from our athlete. Visual comparison between the hip joint contact forces from the participants in our study with those found on the Orthoload database showed a reasonable agreement with our simulations. It should be noted that the hip and knee joint contact force waveforms from the Orthoload database each were from one participant. Differences between our results and the values from Orthoload might be caused by a combination of differences in hip and knee kinematics, bone and muscle morphology, movement execution technique and velocities, additionally to the different methods to obtain the joint contact forces (simulations versus in-vivo measurement). (DOCX)

**S2 Fig. Validation of simulations with Perrone et al. (2023).** Hip resultant joint contact force waveforms of squats (body weight only) of one participant were comparable to single-legged squats obtained from the results of Perrone et al. (2023). The shape of this study's waveforms was similar to the waveforms obtained from our athlete. The differences observed between our absolute values and the results reported by Perrone et al. (2023) may be attributable to several factors: firstly, the execution of single-legged half squats in the study by Perrone et al. (2023) instead of double-legged deep squats as in the present study. Secondly, the differences in the model employed, the movement execution technique, and the velocities may have contributed to the discrepancies in results. (DOCX)

**S3 Fig. Comparison of muscle activations estimated via static optimization and measured using surface electro-myography during the squat movement.** 0% and 100% of the squat cycle represent the upright standing position, while 50% corresponds to the lowest vertical position of the sacrum and thus the deepest point of the squat. Blue lines indicate muscle activations estimated via Static Optimization, and red lines represent muscle activations measured using surface electromyography (EMG). EMG signals were normalized to the peak activation value obtained from Static Optimization for comparability of waveform profiles.
(DOCX)

**S4 Fig. Change of muscle forces throughout the squat cycle with different intensities.** 0% and 100% of the squat cycle represent the respective upright position, and 50% denotes the lowest point of the sacrum, thus representing the deepest squat position. Solid lines in the figures represent the mean joint contact force across participants. Shaded areas represent ± one standard deviation. The lines below the figures indicate significant differences (SPM) during the squat cycle. The first four lines (light blue) represent significant differences between 70% 1-RM and 75% 1-RM, 70% 1-RM and 80% 1-RM, 70% 1-RM and 85% 1-RM, and 70% 1-RM and 90% 1-RM. The next three lines (orange) represent significant differences between 75% 1-RM and 80% 1-RM, 75% 1-RM and 85% 1-RM, and 75% 1-RM and 90% 1-RM. The following two lines (yellow) show significant difference between 80% 1-RM and 85% 1-RM, and 80% 1-RM and 90% 1-RM The last line (purple) shows significant differences between 85% 1-RM and 90% 1-RM.
(DOCX)

**S5 Fig. Muscle segmentation using 3D slicer.** a) Manual annotation of the target structures performed in the "Segment Editor" using the "Paint" function. At least ten slices per muscle were marked in each axis to provide a sufficient basis for the interpolation of the intermediate layers. b) "Fill between Slices" function was used to create a continuous segmentation. c) Where interpolation errors were noted, further slices were incorporated until a complete and anatomically precise segmentation was realised.
(DOCX)

**S1 Table. Percentage differences between the eccentric and concentric joint contact force during squats with increasing intensity (70%, 75%, 80%, 85%, 90% of 1-RM).** 1-RM = 1-repetition maximum.
(DOCX)

**S2 Table. 1-RM values and gender of the participants.** 1-RM = 1-repetition maximum.
(DOCX)

## Acknowledgments

The authors would like to extend our sincere gratitude to the participating athletes of the Austrian Powerlifting Federation for their invaluable support and cooperation in this study.

The authors have no conflict of interest to declare.

## Author contributions

**Conceptualization:** Alexander Pürzel, Hans Kainz.

**Data curation:** Alexander Pürzel.

**Formal analysis:** Willi Koller.

**Investigation:** Alexander Pürzel.

**Methodology:** Paul Kaufmann.

**Resources:** Alexander Pürzel.

**Software:** Willi Koller.

**Supervision:** Arnold Baca, Hans Kainz.

**Validation:** Paul Kaufmann.

**Visualization:** Willi Koller, Paul Kaufmann, Elias Kaj Wallnöfer.

**Writing – original draft:** Alexander Pürzel.

**Writing – review & editing:** Willi Koller, Paul Kaufmann, Elias Kaj Wallnöfer, Arnold Baca, Hans Kainz.

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
