## [Decision Letter · Decision Letter 0]

Dear Dr. Pürzel,

Thank you for submitting your manuscript to PLOS ONE. After careful consideration, we feel that it has merit but does not fully meet PLOS ONE’s publication criteria as it currently stands. Therefore, we invite you to submit a revised version of the manuscript that addresses the points raised during the review process.

I am returning your manuscript with two reviews. The reviewers came to different conclusions about the paper, as you will see. After reading the reviews and looking at the manuscript, I concur with the suggestions made by reviewer #1 and the concerns of reviewer #2 regarding validation of the model in deep flexion. I am sorry I cannot be more positive at the moment.

Note that it will have to go through a second round of review.

Please especially note the following suggestions and give them due consideration.

You must provide more details on the modifications made to the model to incorporate three degrees of freedom at the knee joint or provide a citation to an existing model if used, otherwise your paper cannot be accepted.

Please discuss the limitations in validating your work using an older population of people of knee joint replacement subjects. You cited a study [ref 19], line 52, p. 3, that showed that in deep flexion the JCF values were overpredicted by up to 60% in deep flexion (flexion beyond 60 deg.). Please discuss the implications of that study for the high values found in your work on powerlifters. I find the wording in line 52 to be confusing since “lower” flexion angles can be interpreted differently depending on how flexion angle is reported. Ref [19] found an approximate linear dependence in predicted JCF error with flexion angle for squats done by subject with implants, reaching 60% error in deep flexion of around 80 deg. Given these limitations you should at minimum address these concerns in the Discussion and rephrase the conclusions to qualify your statements about finding high JCF values in terms of the increasing potential errors in deeper flexion.

Please discuss other possible means of providing validation of the model in deeper flexion that could be undertaken in future studies.

Please discuss the use of Static Optimization and its limitations and concerns regarding errors in computing the JCF.

We encourage you to submit your revision within sixty days of the date of this decision.

We look forward to receiving your revised manuscript.

Kind regards,

John Leicester Williams, Ph.D.

Academic Editor

PLOS ONE

Journal Requirements:

Reviewers' comments:

Reviewer's Responses to Questions

**Comments to the Author**

1. Is the manuscript technically sound, and do the data support the conclusions?

Reviewer #1: Yes

Reviewer #2: No

2. Has the statistical analysis been performed appropriately and rigorously?

Reviewer #1: Yes

Reviewer #2: Yes

3. Have the authors made all data underlying the findings in their manuscript fully available?

Reviewer #1: Yes

Reviewer #2: Yes

4. Is the manuscript presented in an intelligible fashion and written in standard English?

Reviewer #1: Yes

Reviewer #2: Yes

Reviewer #1: The authors conducted an excellent study analyzing joint contact forces in the hip, knee, and ankle during squats performed by elite powerlifters at 70% to 90% of their 1-RM. Using 3D motion capture, force plates, musculoskeletal modeling, and MRI, they found that joint contact forces increased with intensity, peaking at over 20 times BW in some cases. These findings offer valuable insights into squat mechanics, performance optimization, and potential injury risks.

The following suggestions may enhance the study and improve clarity for readers:

1. Strengthen the study implications: While the motivation, aim, methods, and results are clearly presented in the abstract, the implications section is brief and lacks impact. Including key points on training adaptations, injury risk, and rehabilitation would help engage readers.

2. Clarify model modification (Page 8, Line 150): The authors mention modifying the musculoskeletal model by incorporating three degrees of freedom at the knee joint. If the modifications were made by the authors, a detailed explanation is needed. Otherwise, a citation should be provided if an existing model was used.

3. Expand MRI volume calculation details: Additional details on the MRI procedure, along with images, would help readers replicate the study and validate the methodology.

4. Validation limitations: Joint contact forces and muscle forces cannot be directly validated, but the authors attempted to validate them using OrthoLoad data. However, OrthoLoad represents an older population with implanted knees, which may affect contact force estimates. An alternative validation method could involve comparing predicted muscle activations from OpenSim with measured EMG signals. Although EMG data was not collected in this study, discussing this limitation and future research directions in the discussion would be beneficial.

5. High joint contact forces and cartilage adaptation: The reported joint contact forces exceeding 20x BW are extreme and could increase cartilage damage risk. The study already discusses possible reasons, but it would be valuable to reference studies that support the idea of cartilage adaptation in powerlifters. Are there any findings suggesting that powerlifters develop stronger cartilage or that training modifications could help protect joint health?

6. Consider discussing the force-length-velocity relationship of muscles to explain why joint contact forces are higher in the concentric phase. This would provide a stronger biomechanical basis for the observed trends.

7. Expand on training implications: Consider elaborating on how these findings impact training optimization, injury prevention, and rehabilitation, especially regarding load management and long-term joint health.

8. Future research scope: Finite element analysis could be used to analyze tissue-level mechanical response, and EMG-informed neuromusculoskeletal modeling in OpenSim may improve force estimations. Discussing these as potential future directions would add value to the study.

9. Improve the conclusion: The conclusion lacks strong recommendations for powerlifters and does not fully address the practical implications of the findings. Consider adding:

• Key findings summary (e.g., higher joint contact forces during the concentric phase).

• Training recommendations (e.g., squat variations, eccentric loading benefits).

• Injury risk considerations (e.g., potential cartilage damage, adaptation strategies).

• Broader application (e.g., relevance to non-elite athletes or rehabilitation).

Reviewer #2: General Comments:

This study used musculoskeletal models to estimate joint contact forces (JCF) at the hip, tibiofemoral, patellofemoral, and ankle joints during squatting with sub-maximum weights in elite powerlifters. The authors found that model-estimated JCF increased with increasing weight for most (but not all) joints and JCF components. Adding weight increased JCF during most parts of the squat except the early-concentric “sticking” phase. They also found JCF to be higher during the concentric (rising) than the eccentric (lowering) phase. These results may be interesting and useful to some in rehabilitation and athletic training settings, because JCF under high external weights may be associated with athletic performance and injury-related joint and tissue stress.

I think the writing quality of this manuscript is fair. Methods were described in detail, statistical analyses appear appropriate, goals, rationales, and logic of the study are clear, discussion of the results are reasonable, and study data are made fully available either in the manuscript or Supporting Information. However, some major concerns regarding soundness of the approach – especially model validation and justification – diminished my confidence in the validity of the results and conclusions. The reported JCF are of much higher magnitudes than prior studies, but model validation only included waveform comparison to sparse cases from a very different population. The use of Static Optimization, which directly govern JCF estimates, was not well justified. Yet despite considerable uncertainties on the validity of these JCF estimates, a large amount of study findings was directly drawn upon the JCF magnitudes. Due to these concerns, I am hesitant to endorse the validity of the study conclusions, thus regretfully recommend against publishing this work in the PLoS ONE journal. I acknowledge that validating JCF in this athletic population may be difficult. However, the authors should at least collect concurrent experimental information like EMG for participant-specific validation. They may also look for in-vitro testing data on JCF under high external loads, with muscle co-contraction considerations in mind.

Despite recommending against publication, I provide additional comments to specific contexts that may help the authors improve their work and a potential future paper. Please see below.

Abstract:

[P2/L24 (Page 2, Line 24)]: No significant change relative to what?

Introduction:

[P3/L52]: Is this accurate? The cited study highlighted that JCF errors tend to be larger at deep knee flexion. Is it known that common musculoskeletal models are often not suited to simulate deep flexions due to less accurate muscle-tendon paths and properties.

[P4/L66]: What movement specifically? Knee rotation, which is essential for knee function, is also a “movement” of the tibia relative to the femur. Did the authors mean anterior translation?

[P4/L74]: I did not understand this. How does quantifying the JCF address the adaptation and health of passive joint structures?

Material and methods:

[P6/L121]: Please clarify that maximum muscle force scaling here was based on MRI data from 6 participants, but the resulting scale factor was applied to all participants. Until later in the Methods, I thought that muscle force was MRI-based for 6 participants, but generic for others.

[P6/L122]: Would the results be confounded by not controlling for equipment types? I think the equipment differences would influence the powerlifting performance of the athletes.

[P7/L131]: Are these 20 markers included in the caption above? If not, where were they placed?

[P8/L170]: Please provide the specific muscle tension value here.

[P9/L174]: Avoid the term “actual value” because MRI-based maximum force is still an estimate.

[P9/L177]: So, all 29 models were scaled using Equation 1 with S = 3 even for the 6 participants with MRI – meaning the F-iso/ss values in Equation 2 were ultimately not used in any model?

[P9/L180]: Can you justify the use of Static Optimization for estimating muscle activation? I am concerned that the squat movement, especially when performed with extreme external weights during powerlifting, would require substantial leg muscle co-contractions that can be poorly represented by static optimization.

[P9/L184]: How did you define a squat cycle (i.e., start and end of squat motion) from your data?

[P9/L188]: Although the JCF waveforms matched instrumented implant data, the reference data had n = 1 from a very different population, and JCF magnitudes were quite different. Because most of the current study findings directly depend upon JCF magnitudes, I question whether this validation is adequate for supporting the results and conclusions of this study.

Results:

[P11/L224]: What was “the observed behavior”?

[P11/L228]: Both Figure 2 and Abstract [P2/L22] noted that the medio-lateral ankle JCF was not higher during the concentric phase than eccentric, which seem to contradict the statement here.

[P12/L235]: I do not see p-values or effect sizes presented anywhere in Table 2. Are they cropped out on the right side of the Table?

[P14/L248]: Do you have an objective definition of the “early concentric phase”, e.g., ##-##% of the movement?

[P14/L258]: I am still confused what these lines represent. For example, why 4 lines for 70% and 1 line for 90%?

[P14/L277]: My observation of Figure 3 differs from this statement – the two peaks appear to be at the mid-eccentric phase (~25-30%) and the end of concentric (~90%).

Discussion:

[P18/L322]: Simply state “differences were <10%”. The authors should be aware that relevance of the small differences may not be ascertained if it fails to exceed the margin of uncertainty.

[P18/L335]: The barbell weight, or the 1-RM, for each participant should probably be reported as study data. The JCF data points presented in Figure 2 are quite variable, which may be directly related to the variability of 1-RM.

[P18/L339]: The 2 BW was for which joint?

[P18/L343]: I cannot concur with this statement. There is no clear evidence that JCF waveforms during powerlifting squats with heavy weights should be the same as body-weight squats by individuals using endo-protheses. Even so, matched waveforms and fulfilling static optimization equations do not readily confirm that these estimated very high JCF magnitudes are valid. Also see my comment at [P9/L188].

[P19/L367]: I do not think n = 29 is a large sample size for this type of biomechanical studies.

Conclusion:

[P20/L378]: What does “decline in peak JCF” mean? What was the purpose of comparing JCF magnitudes across the joints?

Figures and Tables:

[Table 1] Where did the “relative squat performance” metric numbers come from, how were they measured, and what do they mean?

[Figure 1]: If there are markers on the front side of the body, please also include a front view image. Which of these markers were static-only and which used for dynamic motion tracking?

[Figure 2]: Possibly due to the low resolution of the embedded image, I could not visually locate the median values (white circles) and quartiles (darker areas), although they are not essential.

[Table 2]: This Table is cropped out in the submitted document and incompletely displayed.

**Do you want your identity to be public for this peer review?** For information about this choice, including consent withdrawal, please see our Privacy Policy

Reviewer #1: **Yes: ** Rohan Kothurkar

Reviewer #2: No

---

## [Author Response · Author response to Decision Letter 1]

15 May 2025

Dear Editor,

Dear Reviewers,

We greatly appreciate the time, knowledge, and effort you have invested in evaluating our work.

The detailed and constructive comments provided by the reviewers and the editor were invaluable. These suggestions significantly contributed to improving the clarity, scientific rigor, and overall quality of our paper. All feedback was carefully considered, and comprehensive revisions were made to address the concerns raised.

Regarding the main point of the revision – the use of a validated and unchanged model – we would like to emphasize that all simulations were carried out again using the existing Catelli model, which is validated for low flexion angles (Catelli et al., 2019). Importantly, no adjustments or changes were made to the original model.

With these revisions implemented, we are confident that the manuscript now addresses the reviewers’ concerns and meets the high standards required for publication in PLoS One.

For better and easier handling, we have also uploaded the answers to the reviewers below as a document. We kindly ask you to use this. However, all the information is also available here. In the attached tables, you will find our detailed responses to each comment, as well as the revised manuscript with tracked changes.

Thank you again for your guidance and support. We look forward to your feedback on our resubmission.

Warm regards,

Alexander Pürzel

Corresponding author

Comments and Responses:

Editor:

Changes in the manuscript following the editor’s comments are underlined.

Comment 1:

You must provide more details on the modifications made to the model to incorporate three degrees of freedom at the knee joint or provide a citation to an existing model if used, otherwise your paper cannot be accepted.

Answer 1:

Since this point is absolutely central and it is also of utmost importance to us to present comprehensible and valid results, we have carried out all simulations again with the existing Catelli model validated for low flexion angles (Catelli et al., 2019) without any adjustments or changes to the original model. Accordingly, all results, figures, and tables in the revised manuscript are now exclusively based on outputs generated using the unaltered Catelli model. This ensures that our findings are grounded in an established and validated modelling framework.

Comment 2:

Please discuss the limitations in validating your work using an older population of people of knee joint replacement subjects. You cited a study [ref 19], line 52, p. 3, that showed that in deep flexion the JCF values were overpredicted by up to 60% in deep flexion (flexion beyond 60 deg.). Please discuss the implications of that study for the high values found in your work on powerlifters. I find the wording in line 52 to be confusing since “lower” flexion angles can be interpreted differently depending on how flexion angle is reported. Ref [19] found an approximate linear dependence in predicted JCF error with flexion angle for squats done by subject with implants, reaching 60% error in deep flexion of around 80 deg. Given these limitations you should at minimum address these concerns in the Discussion and rephrase the conclusions to qualify your statements about finding high JCF values in terms of the increasing potential errors in deeper flexion.

Answer 2:

We thank the editor for pointing out this important limitation regarding the validation of our model using data from older individuals with knee joint replacements. In response, we have clarified this issue both in the Introduction and in the Limitations in the Discussion Section. We have moved the discussion of this fact from the Introduction to the Discussion, as it is more suitably aligned there.

As a result, we expanded the Limitations in the Discussion Section to address the implications of this limitation for our own results. We now explain that while our validation is based on in vivo data from a less athletic population under lower loading conditions, we used a more recent musculoskeletal model developed after the study by Schellenberg et al., specifically to improve simulation accuracy in deep flexion. This is now clearly stated in the Limitations section. Although the model may reduce the overestimation bias, we acknowledge that the absolute JCF values – particularly in deep flexion – should still be interpreted with caution:

Line 448: Thirdly, the validation of the simulation was based on in vivo data from individuals with instrumented joint replacements [41]. These datasets originate from an older and less athletic population with substantially lower external loads than those observed in our elite powerlifters. Moreover, Schellenberg et al. [78] showed that musculoskeletal models can overestimate tibiofemoral joint contact forces by up to 60 % at deep knee flexion angles (≥80°), which are common in powerlifting squats. However, their findings refer only to models that were available at the time of their study. In our work, we used a musculoskeletal model [28] that was published after Schellenberg et al. [78] and specifically designed to improve accuracy in deep flexion. Although direct in vivo validation in athletic populations remains challenging, this more recent model implementation may reduce the overestimation bias in high flexion angles. Nevertheless, the absolute joint contact force values – particularly in deep flexion – should still be interpreted with caution.

Comment 3:

Please discuss other possible means of providing validation of the model in deeper flexion that could be undertaken in future studies.

Answer 3:

We agree that this is a very important point to discuss.

We fully acknowledge the limitations of using in vivo validation data from individuals with instrumented joint replacements, who typically represent an older, less athletic population. As such, the absolute JCF magnitudes may not be directly transferable to elite powerlifters. To strengthen the validation of our simulation approach, we added two additional efforts:

1. EMG comparison: While full EMG datasets were not usable due to technical limitations, high-quality EMG signals from one participant were available. These EMG-based muscle activations were compared with the muscle activations predicted by OpenSim using Static Optimization. The qualitative agreement between measured and simulated activation patterns serves as an important cross-validation and is now included as an additional figure in the Supporting Information. We have also pointed this out in the methods section.

Line 207: Furthermore, for one participant with available electromyography (EMG) data, we additionally compared the predicted muscle activation patterns of m. biceps femoris, m. gluteus maximus, m. rectus femoris and m. vastus lateralis from OpenSim's Static Optimization with the corresponding surface EMG signals. This qualitative comparison served as an additional validation step for evaluating the plausibility of the simulated neuromuscular control patterns during squatting.

2. New comparison using unloaded squats: We additionally recorded unloaded squats from the same elite powerlifter used in the original simulations and compared the resulting JCF waveforms with those from instrumented prostheses in the OrthoLoad database. We also compared the hip joint contact force waveform between our model and that of Perrone et al. (2023). We have added the validation results in the Result Section:

Line 238: The shape of the hip and knee joint contact force waveforms from our simulations showed a reasonable agreement with the values from instrumented implants (Figure S1 in the Supporting Information) but with higher force magnitudes. Moreover, hip joint contact forces showed similar values and shape compared to previous reported values from a musculoskeletal modelling study [16] (Figure S2 in the Supporting Information). Estimated muscle activations from Static Optimization showed reasonable agreement with experimental EMG signals for all analysed muscles, except the m. rectus femoris (Figure S3 in the Supporting Information).

While the waveforms closely aligned, the absolute JCF values were higher in our athlete. This discrepancy is likely attributable to differences in squat technique (deeper squats, wider stance), individual anthropometrics, and the limitations of using generic musculoskeletal models. These points are now addressed and discussed in depth in large parts of the Discussion and Limitation sections at the end of the Discussion sections.

The Supporting Information was updated with the new JCF-comparisons. Furthermore, we also performed gait analysis in this participant and found JCF magnitudes in line with previous OpenSim studies (Koller et al., 2023), which further supports the physiological plausibility of our model estimates.

Comment 4:

Please discuss the use of Static Optimization and its limitations and concerns regarding errors in computing the JCF.

Answer 4:

Thank you for bringing this very important point to our attention. We have addressed it in the discussion section:

Line 413: While musculoskeletal simulations using Static Optimization offer computational efficiency and simplicity, it has inherent limitations that can affect the accuracy of joint contact force predictions. Static Optimization operates under the assumption that muscle forces can be determined independently at each time point, disregarding the temporal dynamics of muscle activation and contraction [67]. Further, it typically aims to minimise a cost function, such as the sum of squared muscle activations, which may lead to an underestimation of muscle co-contraction. This can result in inaccuracies in joint contact force estimations, particularly in movements where co-contraction plays a role [68]. As illustrated in the validation plot (Figure S3 in the Supporting Information), the activation waveforms estimated via Static Optimization showed a good agreement with surface EMG signals across most muscles. However, activation of the rectus femoris was virtually absent in the simulation, despite clear EMG activity in this muscle. This discrepancy is a known and well-documented limitation of Static Optimization approaches, particularly for biarticular muscles such as the rectus femoris [69, 70, 71]. The minimization-based cost function used in Static Optimization tends to reduce activations that generate opposing joint torques, which can result in physiologically unrealistic suppression of such muscles during tasks involving simultaneous hip and knee extension. Incorporating participant-specific electromyography data in the simulations might enhance the accuracy of muscle force estimations.

Reviewer 1:

Changes in the manuscript following reviewer 1’s comments are highlighted in blue.

Comment 1:

Strengthen the study implications: While the motivation, aim, methods, and results are clearly presented in the abstract, the implications section is brief and lacks impact. Including key points on training adaptations, injury risk, and rehabilitation would help engage readers.

Answer 1:

Thank you for your suggestion to make the abstract more attractive.

We have rewritten large parts of the abstract to stay within the word count guidelines. We have now included a larger section relating to the practical implications of the study results:

Line 20: In contrast to the hip and ankle joints, which exhibited peak resultant joint contact forces in the deepest squat positions, the tibiofemoral and patellofemoral joints maintained consistently high loads over a broad portion of the squat cycle. During large parts of the concentric phase, the resultant joint contact forces did not significantly differ between intensity conditions, with the exception of the hip joint contact force. At 90 % 1-RM, average peak joint contact forces reached 15.5±3.0 times body weight (BW), 23.2±3.9 BW, 26.7±4.3 BW, and 11.5±2.2 BW for the hip, tibiofemoral, patellofemoral, and ankle joint, respectively. The high and sustained joint contact forces observed in our study emphasize the need for load management strategies to optimize performance and reduce injury risk. These insights offer a valuable foundation for tailoring strength training programs and supporting long-term joint health in high-performance athletes.

Comment 2:

Clarify model modification (Page 8, Line 150): The authors mention modifying the musculoskeletal model by incorporating three degrees of freedom at the knee joint. If the modifications were made by the authors, a detailed explanation is needed. Otherwise, a citation should be provided if an existing model was used.

Answer 2:

Since this point is absolutely central and it is also of utmost importance to us to present comprehensible and valid results, we have carried out all simulations again with the existing Catelli model validated for low flexion angles (Catelli et al., 2019) without any adjustments or changes to the original model. Accordingly, all results, figures, and tables in the revised manuscript are now exclusively based on outputs generated using the unaltered Catelli model. This ensures that our findings are grounded in an established and validated modelling framework.

Comment 3:

Expand MRI volume calculation details: Additional details on the MRI procedure, along with images, would help readers replicate the study and validate the methodology.

Answer 3:

We agree. We have included additional important information on reproducibility in the methods section and images in the Supporting Information.

Line 137: Muscle segmentation was performed using the open-source software 3D Slicer (version 5.2.2) [26]. Manual annotations were created with at least ten slices per muscle marked in each anatomical plane to provide a robust basis for interpolation. The remaining layers were interpolated, generating continuous 3D segmentations of the target muscles (mm. vasti, m. rectus femoris, m. semimembranosus, m. gluteus maximus). If interpolation errors were detected, additional slices were manually segmented until an anatomically accurate muscle representation was achieved and final muscle volumes were calculated. For a better illustration of the process, please refer to Figure S5 in the Supporting Information.

Comment 4:

Validation limitations: Joint contact forces and muscle forces cannot be directly validated, but the authors attempted to validate them using OrthoLoad data. However, OrthoLoad represents an older population with implanted knees, which may affect contact force estimates. An alternative validation method could involve comparing predicted muscle activations from OpenSim with measured EMG signals. Although EMG data was not collected in this study, discussing this limitation and future research directions in the discussion would be beneficial.

Answer 4:

Thank you for the comment.

We fully acknowledge the limitations of using in vivo validation data from individuals with instrumented joint replacements, who typically represent an older, less athletic population. As such, the absolute JCF magnitudes may not be directly transferable to elite powerlifters. To strengthen the validation of our simulation approach, we added two additional efforts:

1. EMG comparison: While full EMG datasets were not usable due to technical limitations, high-quality EMG signals from one participant were available. These EMG-based muscle activations were compared with the muscle activations predicted by OpenSim using Static Optimization. The qualitative agreement between measured and simulated activation patterns serves as an important cross-validation and is now included as an additional figure in the Supporting Information. We have also pointed this out in the methods section.

Line 207: Furthermore, for one participant with available electromyography (EMG) data, we additionally compared the predicted muscle activation patterns of m. biceps femoris, m. gluteus maximus, m. rectus femoris and m. vastus lateralis from OpenSim's Static Optimization with the corresponding surface EMG signals. This qualitative comparison served as an additional validation step for evaluating the plausibility of the simulated neuromuscular control patterns during squatting.

2. New compar

---

## [Decision Letter · Decision Letter 1]

Dear Dr. Pürzel,

Thank you for submitting your manuscript to PLOS ONE. After careful consideration, we feel that it has merit but does not fully meet PLOS ONE’s publication criteria as it currently stands. Therefore, we invite you to submit a revised version of the manuscript that addresses the points raised during the review process.

We look forward to receiving your revised manuscript.

Kind regards,

John Leicester Williams, Ph.D.

Academic Editor

PLOS ONE

Journal Requirements:

Reviewers' comments:

Reviewer's Responses to Questions

**Comments to the Author**

Reviewer #1: All comments have been addressed

Reviewer #2: (No Response)

2. Is the manuscript technically sound, and do the data support the conclusions?

Reviewer #1: Yes

Reviewer #2: Yes

3. Has the statistical analysis been performed appropriately and rigorously?

Reviewer #1: Yes

Reviewer #2: Yes

4. Have the authors made all data underlying the findings in their manuscript fully available?

Reviewer #1: Yes

Reviewer #2: Yes

5. Is the manuscript presented in an intelligible fashion and written in standard English?

Reviewer #1: Yes

Reviewer #2: Yes

Reviewer #1: Thank you for your detailed and thoughtful responses to the comments. The revisions have significantly enhanced the manuscript’s clarity, scientific rigor, and practical relevance.

I especially appreciate the expanded discussion on cartilage adaptation, the improved methodological transparency, and the strengthened implications for training and injury prevention. Including EMG-based cross-validation and providing a clear justification for using the validated Catelli model without modifications demonstrate strong scientific integrity. The additional details on MRI volume calculation and the explanation of the force-length-velocity relationship further enrich the manuscript.

Your efforts to make the findings more accessible and applicable to both research and practice are commendable.

Well done.

Reviewer #2: The authors substantially revised this manuscript to address the prior concerns, and I command on their diligent efforts. My primary concern on the original paper was the validity of the models and their high-magnitude JCF estimates. In this revision, the authors added new validation data from one participant during bodyweight squats, compared the JCF estimates to a recent study on this same exercise, and included EMG data from one participant to compare with the model estimates. Even with n = 1, these data still increased my confidence in the models. The authors also emphasized in Discussion the limitations on validation and caution to the result data, which are important and necessary additions. With these, I have been convinced that the authors now have the minimum necessary evidence for the validity of their models to be deemed acceptable. Beyond model validation, the authors also greatly improved the clarity of their paper, and most of my previous concerns on clarity have been resolved. I also agree that the authors' additional discussions in response to the other reviewer will enhance the value of this work. Overall, I am satisfied with the quality of the revised work, and I have no other major concerns. Several minor points of ambiguity or confusion remains that I think should be corrected or explained – please see below. Page [P] and line numbers [L] refer to the revised manuscript with tracked changes.

In the manuscript main text:

[P4/L58]: For clarity, replace “the movement” with “anterior translation”.

[P6/L108]: 73 and 55 do not subtract down to 20 calibration-only markers.

[P9/L181]: While the new descriptions clarified that Equation 2 was used to determine the scale factor S, it is confusing that this is not represented in the equations, leaving Equation 1 and 2 disjointed. Would it be correct to formulate that S ~= mean (F_iso / F_scaled_equation_1) = 3 (with n = 6), then F_scaled_final = F_scaled_equation_1 * S (for n = 29)? Also, please add back the caption that “ss” means subject specific.

[P10/L208]: Were EMG data also acquired from the unloaded squat, or from a sub-1-RM loaded squat? Were the EMG and JCF validation data from the same participant? Please clarify.

[P18/L294]: In the updated Figure 3, it appears that hip JCF is now medial (-), not lateral.

[P21/L358]: Why is the patellofemoral JCF described again here, with a different value from the previous sentence? Please double check, as citation [49] seems to be a study on hip JCF.

[Table 1]: For clarity and consistency, can you simply replace “relative squat performance” with “one-repetition maximum (1-RM)”?

[Figures 2 and 3]: I was able to access the original TIFF figures, and they are of good resolution. What do the shapes of the shaded areas represent? Are these violin plots? Also clarify what the shaded areas around the Figure 3 waveforms represent.

In Supporting Information:

[Figure S1]: Use black color on the right-side Y scale to match the correct data line. Change the yellow line label to “Squat 15kg”.

[Figure S2]: Does each of these represent double- or single-legged squats? I would expect JCFs to approximately double when countering full versus half body weight.

[Figure S3]: Please set all Y scales to 0–1. Model-estimated rectus femoris activation appears on this figure as if a large asymmetry existed, but the finding was that they were both near-zero.

[Figure S4]: Update figure caption to clarify what the waveform shadings and SPM lines mean.

[Table S2]: It may be useful to also report these values as normalized (xBM) in addition to kg.

**Do you want your identity to be public for this peer review?** For information about this choice, including consent withdrawal, please see our Privacy Policy

Reviewer #1: **Yes: ** Rohan Kothurkar

Reviewer #2: No

---

## [Author Response · Author response to Decision Letter 2]

20 Jun 2025

Dear Editor,

Dear Reviewers,

Please allow us to express our sincere gratitude for the considerable time you have dedicated to the initial review, which was tremendously beneficial in enhancing the manuscript. It is our understanding that the constructive and valuable input that has been provided has resulted in a significant enhancement in the quality of the work.

We would also like to express our gratitude for the new feedback, which has helped to make the manuscript even more scientifically rigorous. We have done our utmost to address every single point.

We are pleased to say that, after careful consideration, we are now confident that the manuscript has addressed all the concerns raised and meets the high standards required for publication in PLoS One.

Thank you again for your guidance and support. We look forward to your feedback on our resubmission.

For better and easier handling, we have also uploaded the answers to the reviewers below as a document. We kindly ask you to use this. However, all the information is also available here. In the attached tables, you will find our detailed responses to each comment, as well as the revised manuscript with tracked changes.

Warm regards,

Alexander Pürzel

Corresponding author

Comments and Responses:

Reviewer 1:

General Comment:

Thank you for your detailed and thoughtful responses to the comments. The revisions have significantly enhanced the manuscript’s clarity, scientific rigor, and practical relevance.

I especially appreciate the expanded discussion on cartilage adaptation, the improved methodological transparency, and the strengthened implications for training and injury prevention. Including EMG-based cross-validation and providing a clear justification for using the validated Catelli model without modifications demonstrate strong scientific integrity. The additional details on MRI volume calculation and the explanation of the force-length-velocity relationship further enrich the manuscript.

Your efforts to make the findings more accessible and applicable to both research and practice are commendable.

Well done.

General Response:

We would like to sincerely thank you for your time, effort, and expertise in reviewing our manuscript. Your thoughtful and detailed comments were highly valuable in guiding our revisions. We are grateful for your positive review and pleased that the changes you suggested improved the clarity, scientific quality and practical relevance of the manuscript. Your feedback was both encouraging and motivating.

Thank you again for your support.

Reviewer 2:

Changes in the manuscript following the reviewer’s comments are written in red.

General Comment:

The authors substantially revised this manuscript to address the prior concerns, and I command on their diligent efforts. My primary concern on the original paper was the validity of the models and their high-magnitude JCF estimates. In this revision, the authors added new validation data from one participant during bodyweight squats, compared the JCF estimates to a recent study on this same exercise, and included EMG data from one participant to compare with the model estimates. Even with n = 1, these data still increased my confidence in the models. The authors also emphasized in Discussion the limitations on validation and caution to the result data, which are important and necessary additions. With these, I have been convinced that the authors now have the minimum necessary evidence for the validity of their models to be deemed acceptable. Beyond model validation, the authors also greatly improved the clarity of their paper, and most of my previous concerns on clarity have been resolved. I also agree that the authors' additional discussions in response to the other reviewer will enhance the value of this work. Overall, I am satisfied with the quality of the revised work, and I have no other major concerns. Several minor points of ambiguity or confusion remains that I think should be corrected or explained – please see below. Page [P] and line numbers [L] refer to the revised manuscript with tracked changes.

General Response:

We are very grateful for your thoughtful and constructive feedback, as well as the time and expertise you dedicated to reviewing our revised manuscript. Your recognition of our efforts to improve model validation, address key concerns, and enhance the clarity of the paper is greatly appreciated. We are especially encouraged by your assessment that the revisions have sufficiently addressed your main concerns regarding model validity.

Regarding the remaining minor points of ambiguity or confusion, we carefully reviewed each of them and did our best to clarify and improve these aspects in the manuscript. Thank you again for your valuable input and support throughout the review process.

Comment 1:

For clarity, replace “the movement” with “anterior translation”.

Response 1:

We agree. That is a much more appropriate and understandable expression.

Line 57: Furthermore, compressive forces are of paramount importance for stabilising the knee joint, as they counteract anterior-posterior shear forces and, therefore, reduce the anterior translation of the tibia in relation to the femur while also contributing to frontal plane stability by limiting varus and valgus motion [23].

Comment 2:

73 and 55 do not subtract down to 20 calibration-only markers.

Response 2:

Thank you for pointing out this typo. We have corrected it.

Line 107: A three-dimensional motion capture system comprising 12 cameras (Vicon Motion System, Oxford, UK) was used to capture the trajectories of 73 (static position) and 53 (squat movement) markers on the lower limbs, upper body, and barbell with a recording frequency of 200 Hz (Figure 1).

Comment 3:

While the new descriptions clarified that Equation 2 was used to determine the scale factor S, it is confusing that this is not represented in the equations, leaving Equation 1 and 2 disjointed. Would it be correct to formulate that S ~= mean (F_iso / F_scaled_equation_1) = 3 (with n = 6), then F_scaled_final = F_scaled_equation_1 * S (for n = 29)? Also, please add back the caption that “ss” means subject specific.

Response 3:

Thank you for the advice on making this section clearer and easier to understand. We have taken on board your valuable feedback for the manuscript and revised the section accordingly.

Line 162: In order to ensure that the model is capable of producing necessary internal joint moments to resist the external moments, the maximum isometric muscle forces (F) for all models were adjusted based on their body mass (M) using Equation 1 [30, 31].

Equation 1 F_(iso,scaled)=F_(iso,generic)*(M_scaled/M_generic )^(2/3)

Despite this muscle force scaling process, the simulations resulted in reserve actuator forces that were beyond 10 % of the net joint moments, thereby producing simulations that were not realistic [32]. In other words, the models were too weak to perform the high load exercises of the elite-level athletes. In order to obtain a more precise estimation of the isometric maximum force of the participants and to make the necessary adjustments to the models, MRI scans of the lower limbs were analysed from six participants. The vasti, rectus femoris, semimembranosus, and gluteus maximus muscles were segmented and their volumes were calculated using the 3D Slicer 5.2.2 [26]. The maximum isometric muscle force was subsequently calculated using Equation 2 [33, 34], where F_iso represents the muscle's isometric maximum force, ss means subject specific, V is the muscle volume determined by the MRI scans and l_o^m is the optimal fibre length (OpenSim 4.4 default [27]). The maximum fibre tension (σ) was set to 105 N cm⁻², a value which falls within the range previously reported in the literature [35]. This choice was based on anatomical cross-sectional areas measured in bodybuilders [36] and was adjusted for muscle pennation angle [36, 37, 38] to better reflect the physiological cross-sectional area.

Equation 2 F_(iso,ss)=V_ss/(l_(o,ss)^m )*σ

The maximum isometric force calculated using Equation 1 was found to be three times lower (S obtained with Equation 3) than the corresponding estimate using Equation 2, which was based on MRI-derived muscle volume data.

Equation 3 S≈mean F_(iso,ss)/F_(iso,scaled)

Consequently, the final equation (Equation 4) employed for the scaling of isometric forces across all 29 models involved multiplying the initial equation (Equation 1) by factor S (Equation 3), to ensure that the models are capable of withstanding the external loads.

Equation 4 F_(iso,scaled_final)=F_(iso,generic)*(M_scaled/M_generic )^(2/3)*S

Although Equation 2 was applied only to the muscle volume values of the MRI subgroup (6 athletes), its output was used to define the scaling factor in Equation 4. Given the homogeneity of the participants with respect to their performance level, Equation 4 with S=3 was employed to scale the isometric maximum force for the models of all 29 participants.

Comment 4:

Were EMG data also acquired from the unloaded squat, or from a sub-1-RM loaded squat? Were the EMG and JCF validation data from the same participant? Please clarify.

Response 4:

Thank you for this notice. We have clarified the statement.

Line 212: Furthermore, for the same participant, electromyography (EMG) data were available and used as an additional validation step for evaluating the plausibility of the simulated neuromuscular control patterns during squatting. For the 70 % 1-RM condition, we qualitatively compared EMG signals of m. biceps femoris, m. gluteus maximus, m. rectus femoris, and m. vastus lateralis with muscle activation output from our musculoskeletal simulations (i.e., obtained from Static Optimization).

Comment 5:

In the updated Figure 3, it appears that hip JCF is now medial (-), not lateral.

Response 5:

Thank you. We very much appreciate it, and have changed the wording in the manuscript to “medially”.

Line 300: The medio-lateral forces, which showed only minimal changes with increasing intensities, were small compared to the other force components and had a medially oriented vector.

Comment 6:

Why is the patellofemoral JCF described again here, with a different value from the previous sentence? Please double check, as citation [49] seems to be a study on hip JCF.

Response 6:

Thank you for pointing out this typo. We have corrected it.

Line 364: In the hip joint, forces of approximately 2 BW were observed during squats without additional loads [49].

Comment 7:

For clarity and consistency, can you simply replace “relative squat performance” with “one-repetition maximum (1-RM)”?

Response 7:

We agree and have replaced the term as suggested by the reviewer.

Comment 8:

I was able to access the original TIFF figures, and they are of good resolution. What do the shapes of the shaded areas represent? Are these violin plots? Also clarify what the shaded areas around the Figure 3 waveforms represent.

Response 8:

Thank you for pointing out the missing information. The shaded areas are violin plots representing the distribution of data for each condition using kernel density estimation. The darker shaded sections indicate the interquartile range, while the lighter sections represent the remainder of the distribution. We have updated the figure caption to clarify this.

Line 261: Shaded areas represent violin plots, illustrating the distribution of data; darker areas indicate quartiles, while lighter areas show the remainder of the distribution.

In Figure 3, the shaded areas around each waveform represent ±1 standard deviation. We have added this explanation to the figure caption for clarity.

Line 285: Solid lines in the figures represent the mean joint contact force across participants. Shaded areas represent ± one standard deviation.

Comment 9:

Use black color on the right-side Y scale to match the correct data line. Change the yellow line label to “Squat 15kg”.

Response 9:

We agree. That makes the figure much easier to understand. Thank you very much. We have adapted Figure S1 accordingly.

Comment 10:

Does each of these represent double- or single-legged squats? I would expect JCFs to approximately double when countering full versus half body weight.

Response 10:

Thank you very much for this input. The study by Perrone et al. (2023) involved single-legged squats, our study involved double-legged squats. We have now updated the label on Figure S2 to reflect this. The fact that the participants in the Perrone et al. (2023) study performed only half squats that were not as deep as in our study may explain why the absolute values are lower despite the one-legged performance. Therefore, we have revised the caption to clarify that the Perrone et al. (2023) study involved single-legged half squats, which could explain the lower absolute hip joint contact forces observed.

Caption Figure S2: Hip resultant joint contact force waveforms of squats (body weight only) of one participant were comparable to single-legged squats obtained from the results of Perrone et al. (2023). The shape of this study’s waveforms was similar to the waveforms obtained from our athlete. The differences observed between our absolute values and the results reported by Perrone et al. (2023) may be attributable to several factors: firstly, the execution of single-legged half squats in the study by Perrone et al. (2023) instead of double-legged deep squats as in the present study. Secondly, the differences in the model employed, the movement execution technique, and the velocities may have contributed to the discrepancies in results.

Comment 11:

Please set all Y scales to 0–1. Model-estimated rectus femoris activation appears on this figure as if a large asymmetry existed, but the finding was that they were both near-zero.

Response 11:

We agree. We have adapted Figure S3 accordingly.

Comment 12:

Update figure caption to clarify what the waveform shadings and SPM lines mean.

Response 12:

Thank you for the comment. We updated the figure caption.

Figure S4: 0% and 100% of the squat cycle represent the respective upright position, and 50% denotes the lowest point of the sacrum, thus representing the deepest squat position. Solid lines in the figures represent the mean joint contact force across participants. Shaded areas represent ± one standard deviation. The lines below the figures indicate significant differences (SPM) during the squat cycle. The first four lines (light blue) represent significant differences between 70 % 1-RM and 75 % 1-RM, 70 % 1-RM and 80 % 1-RM, 70 % 1-RM and 85 % 1-RM, and 70 % 1-RM and 90 % 1-RM. The next three lines (orange) represent significant differences between 75 % 1-RM and 80 % 1-RM, 75 % 1-RM and 85 % 1-RM, and 75 % 1-RM and 90 % 1-RM. The following two lines (yellow) show significant difference between 80 % 1-RM and 85 % 1-RM, and 80 % 1-RM and 90 % 1-RM The last line (purple) shows significant differences between 85 % 1-RM and 90 % 1-RM.

Comment 13:

It may be useful to also report these values as normalized (xBM) in addition to kg.

Response 13:

We agree and also reported normalized (xBM) values in addition to kg in the Table S2 of the revised manuscript.

---

## [Decision Letter · Decision Letter 2]

Biomechanical analysis of hip, knee, and ankle joint contact forces during squats in elite powerlifters

PONE-D-25-02546R2

Dear Dr. Pürzel,

We’re pleased to inform you that your manuscript has been judged scientifically suitable for publication and will be formally accepted for publication once it meets all outstanding technical requirements.

Kind regards,

John Leicester Williams, Ph.D.

Academic Editor

PLOS ONE

Additional Editor Comments (optional):

Reviewers' comments:

Reviewer's Responses to Questions

**Comments to the Author**

Reviewer #2: All comments have been addressed

2. Is the manuscript technically sound, and do the data support the conclusions?

Reviewer #2: Yes

3. Has the statistical analysis been performed appropriately and rigorously?

Reviewer #2: Yes

4. Have the authors made all data underlying the findings in their manuscript fully available?

Reviewer #2: Yes

5. Is the manuscript presented in an intelligible fashion and written in standard English?

Reviewer #2: Yes

Reviewer #2: Thank you to the authors for their additional revisions in response to my follow-up comments. All previous points of concern have been addressed. I have no further critiques.

**Do you want your identity to be public for this peer review?** For information about this choice, including consent withdrawal, please see our Privacy Policy

Reviewer #2: No

---

## [Editor Report · Acceptance letter]

PONE-D-25-02546R2

PLOS ONE

Dear Dr. Pürzel,

I'm pleased to inform you that your manuscript has been deemed suitable for publication in PLOS ONE. Congratulations! Your manuscript is now being handed over to our production team.

Kind regards,

on behalf of

Dr. John Leicester Williams

Academic Editor

PLOS ONE